# The Comparative Power of ReLU Networks and Polynomial Kernels in the Presence of Sparse Latent Structure

**Frederic Koehler**[*]
Department of Mathematics
Massachusetts Institute of Technology

**Andrej Risteski** [†]
Department of Mathematics and IDSS
Massachusetts Institute of Technology

## Abstract

There has been a large amount of interest, both in the past and particularly recently, into the relative advantage of different families of universal function approximators, for instance neural networks, polynomials, rational functions, etc. However, current research has focused almost exclusively on understanding this problem in a *worst-case setting*: e.g. characterizing the best $L_1$ or $L_\infty$ approximation in a box (or sometimes, even under an adversarially constructed data distribution.) In this setting many classical tools from approximation theory can be effectively used.

However, in typical applications we expect data to be high dimensional, but structured – so, it would only be important to approximate the desired function well on the relevant part of its domain, e.g. a small manifold on which real input data actually lies. Moreover, even within this domain the desired quality of approximation may not be uniform; for instance in classification problems, the approximation needs to be more accurate near the decision boundary. These issues, to the best of our knowledge, have remain unexplored until now.

With this in mind, we analyze the performance of neural networks and polynomial kernels in a natural regression setting where the data enjoys sparse latent structure, and the labels depend in a simple way on the latent variables. We give an almost-tight theoretical analysis of the performance of both neural networks and polynomials for this problem, as well as verify our theory with simulations. Our results both involve new (complex-analytic) techniques, which may be of independent interest, and show substantial qualitative differences with what is known in the worst-case setting.

## 1 Introduction

The concept of representational power has been always of great interest in machine learning. In part the reason for this is that classes of "universal approximators" abound – e.g. polynomials, radial bases, rational functions, etc. Some of these were known to mathematicians as early as Bernstein and Lebesgue[1] – yet it is apparent that not all such classes perform well empirically.

In recent years, the class of choice is neural networks in tasks as simple as supervised classification, and as complicated as reinforcement learning – inspiring an immense amount of theoretical study. Research has focus on several angles of this question, e.g. comparative power to other classes of functions (Yarotsky, 2017; Safran and Shamir, 2017; Telgarsky, 2017), the role of depth and the importance of architecture (Telgarsky, 2016; Safran and Shamir, 2017; Eldan and Shamir, 2016), and many other topics such as their generalization properties and choice of optimization procedure (Hardt et al., 2016; Zhang et al., 2017; Bartlett et al., 2017).

[*]Email: fkoehler@mit.edu. Research is partially supported by NSF Large CCF-1565235 and Ankur Moitra's David and Lucile Packard Fellowship.

[†]Email: risteski@mit.edu.

[1]Lebesgue made use of the universality of absolute value and hence ReLu – see the introduction of (Newman et al., 1964).

Our results fall in the first category: comparing the relative power of polynomial kernels and ReLU networks – with a significant twist, that makes our results more relevant to real-life settings. The flavor of existing results in this subject is roughly the following: every function in a class $\mathcal{C}_1$ can be approximately represented as a function in a different class $\mathcal{C}_2$, with some blowup in the size/complexity of the function (e.g. degree, number of nodes, depth). The unsatisfying aspect of such results is the "worst-case" way in which the approximation is measured: typically, one picks a domain coarsely relevant for the approximation (e.g. an interval or a box), and considers the $L_\infty, L_2, L_1, \ldots$ norm of the difference between the two functions on this domain. In some of the constructions (e.g. (Eldan and Shamir, 2016; Safran and Shamir, 2017)), the evaluation is even more adversarial: it's the mean-square error over a specially-designed measure.

Instead, in practically relevant settings, it's reasonable to expect that approximating a predictor function well only on some "relevant domain" would suffice, e.g. near the prediction boundary or near a lower-dimensional manifold on which the data lives, as would be the case in settings like images, videos, financial data, etc. A good image classifier need not care about "typical" data points from the $\ell_\infty$-ball, which mostly look like white noise.

The difficulty with the above question is that it's not immediate how to formalize what the "relevant domain" is or how to model the data distribution. We tackle here a particularly simple (but natural) incarnation of this question: namely, when the data distribution has *sparse latent structure*, and all we ask is to predict a linear function of the latent variables based upon (noisy) observations. The assumption of sparsity is very natural in the context of realistic, high-dimensional data: sparsity under the correct choice of basis is essentially the reason that methods such as lossy image compression work well, and it is also the engine behind the entire field of *compressed sensing* (Donoho, 2006).

## 2 OVERVIEW OF RESULTS

We will be considering a regression task where the data has a sparse latent structure. More precisely, we wish to fit pairs of (observables, labels) $(X, Y)$ generated by a (*latent-variable*) process:

- Sample a *latent vector* $Z \in \mathbb{R}^m$ from $\mathcal{H}$, where $\mathcal{H}$ is a distribution over sparse vectors.
- To produce $X \in \mathbb{R}^n$, set $X = AZ + \xi$, where the noise $\xi \sim subG(\sigma^2)$ is a subgaussian random vector with variance proxy $\sigma^2$ (e.g. $N(0, \sigma^2 I)$).
- To produce $Y \in \mathbb{R}$, we set $Y = \langle w, Z \rangle$.

We hope the reader is reminded of classical setups like sparse linear regression, compressive sensing and sparse coding: indeed, this distribution on the data distribution $X$ is standard in these setups. In our setting, we additionally attach a regression task to this data distribution, wherein the labels $Y$ are linearly generated[2] by a predictor $w$ from the *latent* vector $Z$.

Note our interest is slightly different than usual: in the traditional setup, we are interested in the statistical/algorithmic problem of inferring $Z$, given $X$ as input (the former studying the optimal rates of "reconstruction" for $Z$, the latter efficient algorithms for doing so). In particular, we do not typically care about the particular form of the predictor as long as it is efficiently computable.

By contrast, we want to understand how well different subsets of universal approximator families can fit the data points $(X, Y)$. Namely, regardless of the specifics of the training procedure, the end will be an element of some function class like a linear function of a kernel embedding of $X$, or a neural network. Therefore, we ask if these classes are rich enough to reconstruct $Y$ given $X$ accurately (i.e. compared to the Bayes-optimal estimator $\mathbb{E}[Y|X]$): if the answer is negative, then we know our predictor will perform poorly, no matter the training method. We measure the performance of these estimators in the natural[3] distributional sense: expected reconstruction error, $\mathbb{E}[(\hat{Y} - Y)^2]$. Informally, what we will show is the following.

**Theorem** (Informal). *For the problem of predicting $Y$ given $X$ in the generative model for data described above, it holds that:*

---

[2] One could also imagine producing discrete labels by applying a softmax operation. We stick to the regression setting for reasons of technical simplicity and leave generalizing our results to future work.

[3] This is natural because it is the (only) loss which the conditional expectation $\mathbb{E}[Y|X]$ minimizes. In statistical language, we are asking about the *minimum excess risk* achievable by estimators in our function class.

*(1) Small two-layer ReLU networks achieve close to the statistically optimal rate.*
*(2) Polynomial predictors of degree lower than* $\log m$ *achieve a statistical rate which is substantially worse. (In fact, in a certain sense, close to "trivial".) Conversely, polynomial predictors of degree* $O((\log n)^2)$ *achieve close to the statistically optimal rate.*

The lower bound in (2) is relevant since fitting a polynomial to data points of the form $(x_i, y_i)$ requires[4] searching through the space of multivariate polynomials of degree $\Omega(\log m)$ which has dimension $m^{\Omega(\log(m))}$, and thus even writing down all of the variables in this optimization problem takes super-polynomial time. Practical aspects of using polynomial kernels even with much lower degree than this have been an important concern and topic of empirical research; see for example (Chang et al., 2010) and references within. On the other hand, the upper bound in (2) shows that our analysis is essentially tight: greater than $polylog(m)$ degree is not required to achieve good statistical performance, which is qualitatively different from the situation in worst-case analyses (see Section 4.2.2 for more details). Our mathematical analysis closely matches the observed behavior in experiments: see Section 6.

For formal statements of the theorems, see Section 4.

## 3 PRIOR WORK

There has been a large body of work studying the ability of neural networks to approximate polynomials and various classes of well-behaved functions, such as recent work (Yarotsky, 2017; Safran and Shamir, 2017; Telgarsky, 2017; Poggio et al., 2017). These results exclusively focus on the worst-case setting where the goal is to find a network close to some function in some norm (e.g. $L_\infty$ or $L_1$-norm, often under an adversarially chosen measure).

In contrast there is little work on the problem of approximating ReLU networks by polynomials, mostly because it is well-known by classical results of approximation theory (Newman et al., 1964; DeVore and Lorentz, 1993) that polynomials of degree $\Omega(1/\epsilon)$ are *required* to approximate even a single ReLU function within error $\epsilon$ in $L_\infty$-norm on $[-1, 1]$. On the other hand, we will show that if we do not seek to achieve $\epsilon$-error everywhere for the ReLU (in particular not near the non-smooth point at 0) we can build good approximations to ReLU using polynomials of degree only $O(\log^2(1/\epsilon))$ (see discussion in Section 4.2.2 and Theorem 5.2).

Because of the trivial $\Omega(1/\epsilon)$ lower bound for worst-case approximation of ReLU networks by polynomials, (Telgarsky, 2017) studied the related problem of approximating a neural network by rational functions. (A classical result of approximation theory (Newman et al., 1964) shows that rational functions of degree $O(\log^2(1/\epsilon))$ can get within $\epsilon$-error of the absolute value function.) In particular, (Telgarsky, 2017) shows that rational functions of degree $polylog(1/\epsilon)$ can get within $\epsilon$ distance in $L_\infty$-norm of bounded depth ReLU neural networks.

Somewhat related is also the work of (Livni et al., 2014) who considered neural networks with quadratic activations and related their expressivity to that of sigmoidal networks in the depth-2 case building on results of (Shalev-Shwartz et al., 2011) for approximating sigmoids. The result in (Shalev-Shwartz et al., 2011) is also proved using complex-analytic tools, though the details are substantially different.

The work of (Zhang et al., 2016) studied the power of kernel regression methods to simulate a certain class of neural networks. More precisely, they bounded the $\ell_2$ norm of kernel regression models approximating neural networks with bounded depth, "nice" activation functions (not including ReLU), and small input and edge weights. By standard generalization theory, this gives a corresponding sample complexity result for improper learning via kernels. In our setting, their result does not apply: first, the network of interest has ReLU activations; even ignoring this issue, their bounds would be

---

[4]We note that for kernel ridge regression, one can use the *kernel trick* to reduce to train in a space with dimension equal to the size of the training dataset. However, since training data sets are often very large, this is not necessarily helpful. More usefully, in the *classification* context, the optimization may become feasible due to the existence of a smaller number of *support vectors*. In this case, the ability of algorithms to efficiently find a good function depends on more complex interactions between the particular choice of kernel and the margin properties of the data; we leave analyzing these kinds of quantities for future work.

roughly exponential in $n$ because the $\ell_2$ norm of the network's input vector is large, of order $\Theta(\sigma\sqrt{n})$.[5].

There is a vast literature on high dimensional regression and compressed sensing which we do not attempt to survey, since the main goal of our paper is *not* to develop new techniques for sparse regression but rather to analyze the representation power of kernel methods and neural networks. Some relevant references for sparse recovery can be found in (Vershynin, 2018; Rigollet, 2017). We only emphasize that the upper bound via soft thresholding we show (Theorem 4.1) is implicit in the literature on high-dimensional statistics; we include the proofs here solely for completeness.

## 4 MAIN RESULTS

In this section we will give formal statements of the results and give some insight into the techniques used.

First, we state the assumptions on the parameters of our generative model:

- $Z$ is sparse: more precisely, $|\mathrm{supp}(Z)| \leq k$ and $\|Z\|_1 \leq M$ with high probability.[6]
- $A$ is a $\mu$-*incoherent* $n \times m$ matrix, which means that $\|A^\top A - I\|_\infty \leq \mu$ for some $\mu \geq 0$.
- $\|w\|_\infty = 1$ (w.l.o.g., since changing the magnitude of $w$ rescales $Y$)

The assumption on $A$ is standard in the literature on sparse recovery (see reference texts (Rigollet, 2017; Moitra, 2018)). In general one needs an assumption like this (or a stronger one, such as the RIP property) in order to guarantee that standard algorithms such as LASSO actually work for sparse recovery. For the reader not familiar with this literature, this property is a proxy for the matrix being "random-like" – e.g. a matrix with i.i.d. entries of the form $\pm 1/\sqrt{n}$ has $\mu = O(1/\sqrt{n})$, even when $m >> n$. We also note that for notational convenience, we will denote $\|A\|_\infty = \max_{i,j} |A_{i,j}|$.

Before proceeding to the results, we note that **the first-time reader may freely assume that $\mu = 0$ and $n = m$**; the results are still interesting in this setting and no important technical idea is needed for the more general case. For the upper bounds, we have included results for the more general setting (with $\mu \geq 0$) to show that our results are relevant even to very high-dimensional settings where $m >> n$. We have only proven the lower bound in the case $\mu = 0$: this is the *easiest* setting for algorithms, so this makes the lower bounds the strongest.

### 4.1 REGRESSION USING RELU NETWORKS

We prove the following theorem, which shows that small 2-layer ReLU networks can achieve an almost optimal statistical rate. Let us denote the soft threshold function with threshold $\tau$ as $\rho_\tau(x) := \mathrm{sgn}(x)\min(0, |x| - \tau) = \mathrm{ReLU}(x - \tau) - \mathrm{ReLU}(-x + \tau)$. Let's introduce the notation $\rho_\tau^{\otimes m}$ to denote the map given by applying $\rho_\tau$ coordinate-wise to a vector in $\mathbb{R}^m$. Consider the following estimator (for $y$), corresponding to a 2-layer neural network:

$$\hat{Z}_{NN} := \rho_\tau^{\otimes m}(A^\top X)$$

$$\hat{Y}_{NN} := \langle w, \hat{Z}_{NN} \rangle$$

We can prove the following result for the estimator (see Appendix A of the supplement):

**Theorem 4.1** (2-layer ReLU). *With high probability, the estimator $\hat{Y}_{NN}$ satisfies*

$$(\hat{Y}_{NN} - Y)^2 = O((1 + \mu)\sigma^2 k^2 \log(m) + \mu^2 k^2 M^2)$$

Notice that the size of the ReLU net is comparable to the input: one of the layers has the same dimension as $A$, the other the same dimension as $w$. Furthermore, to interpret this result, recall that we think of $\mu$ as quite small – in particular $\mu \ll 1$. Thus the error of the estimator is essentially $O(\sigma^2 k^2 \log(m))$, i.e. essentially $|\sigma|$ error "per-nonzero-coordinate". It can be shown that this upper

---

[5]See Proposition 1 of Zhang et al. (2016)

[6]The assumed 1-norm bound $M$ plays a minor role in our bounds and is only used when the incoherence $\mu > 0$.

bound is nearly information-theoretically optimal (see Remark B.1), except that there is an additional factor of $k$. This additional factor is artificial and can be removed with added technical effort; we show how to do this in the $\mu = 0$ case in Theorem A.1.

We emphasize that the analysis of this kind of soft thresholding estimator is implicit in much of the literature on sparse linear regression. For completeness, we include a complete and self-contained proof of Theorem 4.1 in Section A.

## 4.2 REGRESSION USING POLYNOMIALS

### 4.2.1 LOWER BOUNDS FOR LOW-DEGREE POLYNOMIALS

We first show that polynomials of degree smaller than $O(\log m)$ essentially cannot achieve a "non-trivial" statistical rate. This holds even in the easiest case for the dictionary $A$: when it's the identity matrix.

More precisely, we consider the situation in which $A$ is an orthogonal matrix (i.e. $\mu = 0, m = n$), $w \in \{\pm 1\}^m$, the noise distribution is Gaussian $N(0, \sigma^2 I)$, and the entries of $Z$ are independently 0 with probability $1 - k/m$ and $N(0, \gamma^2)$ with probability $k/m$. Then we show

**Theorem 4.2.** *Suppose $k < m/2$ and $f$ is a multivariate degree $d$ polynomial. Then*

$$\mathbb{E}[(f(X) - Y)^2] \geq (1/4) \frac{\gamma^2 k}{\left(1 + \sqrt{k/m}(d+1)^{3d+2} \left(1 + (\gamma/\sigma)^d\right)\right)^2}$$

To parse the result, observe that the numerator is of order $\gamma^2 k$ which is the error of the trivial estimator[7] and the denominator is close to 1 unless $d$ is sufficiently large with respect to $m$. More precisely, assuming the signal-to-noise ratio $\gamma/\sigma$ does not grow too quickly with respect to $m$, we see that the denominator is close to 1 unless $d^d = \Omega(\sqrt{m})$, i.e. unless $d$ is of size $\Omega((\log m)/\log\log m)$. On a technical note we observe that this statement is given with respect to expectation but a similar one can be made with high probability, see Remark B.2.

### 4.2.2 NEARLY MATCHING UPPER BOUNDS

The lower bound of the previous section leaves open the possibility that polynomials of degree $O(\text{polylog}(m))$ still do not suffice to perform sparse regression and solve our inference problem; Indeed, it is a well-known fact (see e.g. (Telgarsky, 2017)) that to approximate a single ReLU to $\epsilon$-closeness in infinity norm in $[-1, 1]$ requires polynomials of degree $\text{poly}(1/\epsilon)$; this follows from standard facts in approximation theory (DeVore and Lorentz, 1993) since ReLU is not a smooth function.

Proceeding with this "worst-case" way of thinking: our upper bound follows by designing a polynomial approximation to ReLU into our neural network construction; since estimates for $Y$ typically accumulate error from estimating each of the $m$ coordinates of $Z$, to guarantee accurate reconstruction we would need $m\epsilon$ to be small. Plugging in the the *best* approximation to ReLU in infinity norm, we would need a $\Omega(\sqrt{m})$-degree polynomial for this to yield a multivariate polynomial with similar statistical performance to the 2-layer ReLU network which computes $\hat{Y}_{NN}$. Thus, naively, we might suspect that the degree of the kernel needs to be as high as $\sqrt{m}$ to get a reasonable approximation.

Surprisingly, we show this intuition is incorrect! In fact, we show how using only a $\text{polylog}(m)$ degree polynomial, our converted ReLU network has similar statistical performance. Formally this is summarized by the following theorem, where $\hat{Y}_{d,M}$ is the corresponding version of $\hat{Y}_{NN}$ formed by replacing each ReLU by our polynomial approximation.

**Theorem 4.3.** *Suppose $\tau = \Theta(\sigma\sqrt{(1+\mu)\log m} + \mu M)$ and $d \geq d_0 = \Omega((2 + \frac{M}{\tau})\log^2(Mm/\tau^2))$. With high probability, the estimator $\hat{Y}_{d,M}$ satisfies[8]*

$$(\hat{Y}_{d,M} - Y)^2 = O(k^2((1+\mu)\sigma^2\log(m) + \mu^2 M^2))$$

---

[7] I.e. the estimator which always returns 0, without looking at the data.

[8] As in Theorem 4.1, there is a spurious factor of $k$ in this bound which can be removed with additional technical effort. In particular in the $\mu = 0$ case we can remove it using the same argument as Theorem A.1; details are omitted.

The idea behind our construction is described in Section 5.3. Our methods are novel and may be of independent interest; we are not aware of a way to get this result using only generic techniques such as FT-Mollification (Diakonikolas et al., 2010).

# 5 OVERVIEW OF PROOFS

In this section, we will sketch the ideas behind the proofs of our results. The full proofs are relegated to the appropriate appendices. We proceed with each of our results in turn.

## 5.1 UPPER BOUND FOR ReLU NETWORKS

As previously mentioned, this kind of result is well-known in the literature on sparse regression and we include a proof primarily for completeness. The intuition is simple: the estimator $\hat{Z}_{NN}$ can make use of the non-linearity in the soft threshold to zero out the coordinates in the estimate $A^\top X$ which are small and thus "reliably" not in the support of the true $z$. Thus, the estimator only makes mistakes on the non-zero coordinates. The full proofs are in Section A.

## 5.2 LOWER BOUND: PROOF SKETCH OF THEOREM 4.2

The proof of Theorem 4.2 has two main ideas, which we detail below:
(1) A structural lemma, which shows that the optimal predictor has a "decoupled" structure along the coordinates of the latent variable.
(2) An analysis of this decoupled estimator using a bias-variance calculation in an appropriately chosen basis.
The full proofs of this Section are in Appendix B.

### 5.2.1 STRUCTURE OF THE OPTIMAL ESTIMATOR

As explained above, our structural lemma shows that the optimal low-degree polynomial estimator decouples along the coordinates of the latent variable. In order to understand why this should be true, first observe that the optimal estimator for $Y = \langle w, Z \rangle$ given $X$ has a particularly simple structure. Concretely, the optimal estimator is the conditional expectation $\mathbb{E}[\langle w, Z \rangle | X] = \sum_i w_i \mathbb{E}[Z_i | X]$, so the optimal estimator for $Y$ simply reconstructs $Z$ as well as possible coordinate-wise, then takes an inner product with $w$.

With this in mind, note the coordinates of $Z$ are independent in our setting, so optimal estimation of $Z_i$ should not depend in any way on reconstructing $Z_j$ for $j \neq i$. This allows us to show that the optimal polynomial of degree $d$ to estimate $Y$ has no "mixed monomials" in an appropriate basis. This is the content of the next lemma, whose proof is in Appendix B.

**Lemma 5.1.** *Suppose $X = AZ + \xi$ where $A$ is an orthogonal $m \times m$ matrix, $Z$ has independent entries and $\xi \sim N(0, \sigma^2 Id)$. Then there exists a unique minimizer $f_d^*$ over all degree $d$ polynomials $f_d$ of the square-loss,*

$$\mathbb{E}[(f_d(A^\top X) - \langle w, Z \rangle)^2]$$

*and furthermore $f_d^*$ has no mixed monomials. In other words, we can write $f_d^*(A^\top X) = \sum_i f_{d,i}^*((A^\top X)_i)$ where each of the $f_{d,i}^*$ are univariate degree $d$ polynomials.*

### 5.2.2 FOURIER ANALYSIS AND BIAS-VARIANCE TRADE-OFF

Once we have reduced to considering estimators with decoupled structure, it becomes feasible to analyze the performance of all possible low degree polynomials using a bias-variance calculation in a carefully chosen basis. This is the second (and more involved) step in the proof. In order to perform the calculation, we need to apply Fourier analytic methods, so we need to switch to an orthonormal basis. Since the noise we chose for the lower bound instance is Gaussian[9], a natural choice is the Hermite polynomials.

---

[9]The proof does not rely heavily on this choice; as long as we choose the correct orthogonal basis of polynomials, the proof would go through with minor modifications.

We review the definition of the Hermite polynomials in Appendix B, but for the purposes of this proof overview, the Hermite polynomials are polynomials $H_{\mathbf{n}}(x)$ indexed by multi-indices $\mathbf{n} \in \mathbb{N}_0^m$ with the important property that they are orthogonal with respect to the standard m-variate Gaussian distribution, namely

$$\mathbb{E}_{X \sim \mathcal{N}(0, \sigma^2 I)} H_{\mathbf{n}}(X/\sigma) H_{\mathbf{n}'}(X/\sigma) = \begin{cases} 0, & \text{if } \mathbf{n} \neq \mathbf{n}' \\ 1, & \text{otherwise} \end{cases}$$

From this, we can derive Plancherel's Theorem in this basis:

**Theorem 5.1** (Plancherel's Theorem in Hermite Basis). *Let* $f(x) = \sum_{\mathbf{n}} \widehat{f}(\mathbf{n}) H_{\mathbf{n}}(x/\sigma)$, *then*

$$\mathbb{E}_{X \sim \mathcal{N}(0, \sigma^2 I)}[|f(X)|^2] = \sum_{\mathbf{n}} |\widehat{f}(\mathbf{n})|^2$$

We use this theorem, along with the structural Lemma 5.1 to perform a bias-variance tradeoff analysis of any predictor: namely, we show
(1) If the Fourier coefficients $|\widehat{f}(\mathbf{n})|$ are large, then the estimator will be very sensitive to noise (i.e. has too high of a *variance*).
(2) On the other hand, if $|\widehat{f}(\mathbf{n})|$ is small and $f$ is low-degree, then the estimator cannot match the correct mean well regardless of noise (i.e. has too high of a *bias*).

Efficient application of Plancherel's theorem is key to proving both results: in the first case, we apply it over the randomness in the noise $\xi$, and in the second case, we apply it over the randomness in the latent vector $Z$, which has Gaussian entries conditioned on its support. Note that when $f$ is sufficiently high-degree, it can effectively take advantage of the difference in scales between the noise and the signal to achieve both low bias and low variance simultaneously: see the following upper bound section for details.

## 5.3 Upper Bound: Proof Sketch of Theorem 4.3

As previously mentioned, it's a result from classical approximation theory that no low-degree polynomial is close to the ReLU function on all of $[-1, 1]$. The crux of these results is that it's hard to approximate ReLU well at 0, its point of non-smoothness.

However, in our setting *precisely* approximating ReLU everywhere is not important for getting a good regression rate: instead, the approximation needs to be very close to 0 when the input is negative, and only *very coarsely* accurate otherwise. The reason for this is the intuition we described for 2-layer ReLU networks: the property of ReLU that is useful in this setting is it's "denoising" ability – the fact that it zeroes out negative inputs.

Consequently, we design a polynomial approximation to ReLU of degree $O(\log^2 n)$ which sacrifices accuracy near the point of non-smoothness in favor of closeness to 0 in the negative region.

More precisely, we prove the following theorem, in which the parameter $\tau$ in our theorem controls the trade-off between the polynomial $p_d$ being close to 0 for $x < 0$ and being close to $x$ for $x > 0$.

**Theorem 5.2.** *Suppose $R > 0$, $0 < \tau < 1/2$ and $d \geq 7$. Then there exists a polynomial $p_d = p_{d,\tau,R}$ of degree $d$ such that for $x \in [-R, 0]$*

$$|p_d(x) - ReLU(x)| \leq 14R\sqrt{\frac{d}{\tau\pi}} e^{-\sqrt{\pi\tau d/4}}$$

*and for $x \in [0, R]$,*

$$|p_d(x) - ReLU(x)| \leq 2R\tau + 2R\sqrt{\frac{4\tau}{\pi d}} + 12R\sqrt{\frac{d}{\tau\pi}} e^{-\sqrt{\pi\tau d/4}}.$$

The proof of this theorem proceeds in two steps:
(1) First, one takes a "soft-max" mollification of ReLU of the form $g_\beta(x) := \frac{1}{\beta} \log(1 + e^{\beta x})$ with an appropriately tuned $\beta$, so that $g_\beta$ is sufficiently close to ReLU.
(2) Second, if $\beta$ is not too large, we prove that the poles (in the complex plane) of the function $g_\beta$ are

| Dimension ($n$) | 32 | 45 | 64 | 91 | 128 |
|---|---|---|---|---|---|
| 2-Layer ReLU Network | 6.500282 | 6.969625 | 7.479449 | 8.324109 | 8.969790 |
| Degree 17 Polynomial | 7.258900 | 7.723297 | 8.727798 | 9.993010 | 10.256913 |

Table 1: Test errors of baseline ReLU network (Section 4.1) and Degree 17 polynomial kernel; error is unnormalized. Experiments were run for $n$ up to $4096$ and the error between the two methods continued to be similar – results are omitted for concision.

not too close to the origin. This, it turns out, governs the polynomial approximability of $g_\beta$ due to a powerful theorem of Bernstein in complex analysis. (See Theorem C.1, and it's quantitative analogue we prove as Theorem C.2.) Once we have this approximation to ReLU, we directly plug it into our 2-Layer ReLU network estimator from Section 4.1 to prove Theorem 4.3.

The full proofs are in Appendix C.

## 6 SIMULATIONS

Finally, we provide synthetic experiments to verify the predictions from Theorem 4.2 and Theorem 4.3. The setup is as follows: we generate a large synthetic data set (with $n = m$ and $\mu = 0$) in the following fashion:

- $A$ is a random orthogonal matrix and $w$ is sampled from a $n$-dimensional standard Gaussian.
- $Z \in \mathbb{R}^n$ is sampled by including each coordinate with probability $k/n$, and sampling a standard Gaussian for each included coordinate.
- $X$ and $Y$ are sampled according to the generative model in Section 2, using Gaussian noise with standard deviation $\sigma$.

For each fixed degree, we fit a polynomial using least-squares regression, and evaluate the performance on a corresponding test set[10] generated in the same fashion (reusing the same $A$ and $w$). Solving the regression problem for large degrees is intractable using standard training methods; to overcome this issue, we used structural observation in Lemma 5.1 to reduce the regression problem for estimating $Y$ from $X$ to that of estimating $Z_i$ given $X_i$, which is a much lower dimensional problem. [11]

The results of the experiment are in Figure 1, graphed on a log scale. All experiments were run with $k = 5$ and $\sigma = 0.06$. We see that for low degrees, i.e. before our prediction error is close to the information-theoretic limit, the log-error decays roughly linearly with respect to polynomial degree. This matches the prediction of the lower bound in Theorem 4.2 after taking a log of the right-hand-side.

For completeness, we also evaluate the baseline 2-Layer ReLU network described in Section 4.1 in the same experimental setup. Table 1 shows the test error of the baseline 2-Layer ReLU network and, for comparison, the best polynomial of degree 17 in the same experiment. Despite the high degree, the ReLU network is still slightly better.

## 7 CONCLUSIONS

In this paper, we considered the problem of providing representation lower and upper bounds for different classes of universal approximators in a natural statistical setup that exhibits sparse latent structure. We hope this will inspire researchers to move beyond the worst-case setup when considering the representational power of different function classes.

---

[10]Small technical remark: our upper bound (Theorem 4.3) is with-high-probability and not in expectation, so we drop the outlier 1 percent of test results which had largest error. The reason is that we may draw a rare tail event where the input is unusually large/non-sparse and then the polynomial predictor may make an exponentially large error in the degree.

[11]Another technical remark: we only run the experiment for odd degrees, since the optimal estimator is an odd function.

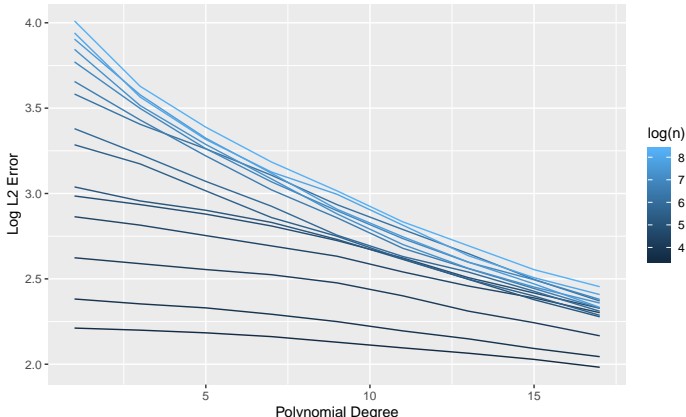

Figure 1: Degree vs Log L2 Error on test set for different values of $n$, the dimensionality of the problem. This plot was generated using a training set of 8000 examples from the generative model and a test set of 1000 additional examples; error is unnormalized.

The techniques we develop are interesting in their own right: unlike standard approximation theory setups, we need to design polynomials which may only need to be accurate in certain regions. Conceivably, in classification setups, similar wisdom may be helpful: the approximator needs to only be accurate near the decision boundary.

Finally, we conclude with a tantalizing open problem: In general it is possible to obtain non-trivial sparse recovery guarantees for LASSO even when the sparsity $k$ is nearly of the same order as $n$ under assumptions such as RIP. Since LASSO can be computed quickly using iterated soft thresholding (ISTA and FISTA, see Beck and Teboulle (2009)), we see that sufficiently deep neural networks can compute a near-optimal solution in this setting as well. It would be interesting to determine whether shallower networks and polynomials of degree polylog($n$) can achieve a similar guarantees.

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

## A  UPPER BOUND FOR 2-LAYER RELU NETWORKS

We will first prove a bound on the error of the soft-thresholding estimator $\hat{Z}_{NN}$ (Lemma A.2), which corresponds to the hidden layer of the neural network: this is essentially a standard fact in high-dimensional statistics (see reference text (Rigollet, 2017)). The idea is that the soft thresholding will correctly zero-out most of the coordinates in the support while adding only a small additional error to the coordinates outside the support.

From the recovery guarantee for $\hat{Z}_{NN}$, we will then deduce Theorem 4.1.

Towards proving the above result, we first need an estimate on the bias of $A^\top x$, i.e. the error without noise:

**Lemma A.1.** *Suppose A is $\mu$-incoherent i.e. $\|A^\top A - Id\|_\infty \le \mu$. Then for any z, $\|A^\top A z - z\|_\infty \le \mu \|z\|_1$.*

*Proof.* We have

$$(A^\top A z)_i = \langle A_i, \sum_j z_j A_j \rangle = z_i \langle A_i, A_i \rangle + \sum_{j \ne i} z_j \langle A_i, A_j \rangle$$

so applying the incoherence assumption we have $|(A^\top A z)_i - z_i| \le \mu \|z\|_1$. □

Using this we can analyze the error in thresholding.

**Lemma A.2.** *Suppose $A$ is $\mu$-incoherent i.e. $\|A^\top A - I\|_\infty \leq \mu$. Let $z$ be an arbitrary fixed vector such that $\|z\|_1 \leq M$ and $|supp(z)| \leq k$. Suppose $x = Az + \xi$ where $\xi \sim N(0, \sigma^2 I_{n \times n})$. Then for some $\tau = \Theta(\sigma\sqrt{(1 + \mu)\log m} + \mu M)$ and $\hat{z} = \rho_\tau^{\otimes n}(A^\top x)$, with high probability we have $\|\hat{z} - z\|_\infty \leq 2\tau$ and $supp(\hat{z}) \subset supp(z)$.*

*Proof.* Observe that
$$A^\top x = z + (A^\top A - I)z + A^\top \xi.$$
Note that entry $i$ of $A^\top \xi$ is $\langle A_i, \xi \rangle$ where $\|A_i\|_2^2 \leq (1 + \mu)$ so $(A^t\xi)_i$ is subgaussian with variance proxy at most $\sigma^2(1 + \mu)$.

By concentration and union bound, with high probability all coordinates not in the true support are thresholded to 0. Similarly we see that for each of the coordinates in the support, an error of at most $2\tau$ is made. $\qquad\square$

From the above lemma, we can easily prove the main theorem of this section:

*Proof of Theorem 4.1.* When the high probability above event happens, we have the following upper bound by Holder's inequality:
$$|\hat{Y}_{NN} - Y|^2 = \langle w|_{\text{supp}(h)}, (\hat{Z}_{NN} - Z)|_{\text{supp}(h)} \rangle^2 \leq k^2 \|\hat{Z}_{NN} - Z\|_\infty^2 = O(k^2((1+\mu)\sigma^2 \log(m) + \mu^2 M^2))$$
$$\square$$

For the lower bounds we will be interested mostly in the case when $\mu = 0$, i.e. $A$ is orthogonal and so $m = n$, the coordinates of $Z$ are independent and each is nonzero with probability at most $k/n$, and the noise is Gaussian. Then the error estimate we had in the previous theorem specializes to $O(\sigma^2 k^2 \log(n))$, but under these assumptions we know that the information-theoretic optimal is actually $\sigma^2 k \log(n)$. While not very important to the flow of the paper, for completeness we can improve the analysis to eliminate the extra factor of $k$, without changing the algorithm:

**Theorem A.1.** *Suppose $A$ is orthogonal (hence $m = n$), the coordinates of $Z$ are independent, and $\xi \sim N(0, \sigma^2 I)$. Then*
$$\mathbb{E}|\hat{Y}_{NN} - Y|^2 = O(k\sigma^2 \log(m))$$

*Proof.* In this case, we have $A^\top X = Z + \xi'$ where $\xi' \sim N(0, \sigma^2 I)$. Therefore the coordinates of $\hat{Z}$ are independent of each other, and so we see
$$\mathbb{E}|\hat{Y}_{NN} - Y|^2 = \sum_i w_i^2 \mathbb{E}[(\hat{Z}_{NN} - Z)_i^2] \leq \sum_i \mathbb{E}[(\hat{Z}_{NN} - Z)_i^2].$$
Let $\mathcal{E}_i$ denote the event that $|\xi'|_i > \tau$. Then
$$\sum_i \mathbb{E}[(\hat{Z}_{NN} - Z)_i^2] = \sum_i \mathbb{E}[(\mathbb{1}_{\mathcal{E}_i} + \mathbb{1}_{\mathcal{E}_i^C})(\hat{Z}_{NN} - Z)_i^2]$$
$$\leq 4k\tau^2 + \sum_i \mathbb{E}[\mathbb{1}_{\mathcal{E}_i^C}(\hat{Z}_{NN} - Z)_i^2]$$
$$= 4k\tau^2 + \sum_i \Pr(\mathbb{1}_{\mathcal{E}_i^C})\mathbb{E}[(\hat{Z}_{NN} - Z)_i^2 | \mathbb{1}_{\mathcal{E}_i^C} = 1]$$
$$\leq 4k\tau^2 + \sum_i \Pr(\mathbb{1}_{\mathcal{E}_i^C})\mathbb{E}[(\tau + |X_i' - Z_i|)^2 | \mathbb{1}_{\mathcal{E}_i^C} = 1]$$
$$\leq 4k\tau^2 + \sum_i \Pr(\mathbb{1}_{\mathcal{E}_i^C})(2\tau^2 + 2\mathbb{E}[|\xi_i'|^2 | \mathbb{1}_{\mathcal{E}_i^C} = 1])$$
$$\leq 4k\tau^2 + \sum_i \frac{C}{m}(2\tau^2 + 2C'\tau^2)$$
where the first inequality follows as in Lemma A.2, the second inequality uses that $|\rho_\tau(x) - x| \leq \tau$, the third uses that $(a + b)^2 = a^2 + 2ab + b^2 \leq 2a^2 + 2b^2$ by Young's inequality, and the last inequality follows from standard tail bounds on Gaussians. We see the last expression is $O(k\sigma^2 \log(m))$ so we have proved the result. $\qquad\square$

# B    LOWER BOUNDS FOR POLYNOMIAL KERNELS

In this section, prove the lower bounds for polynomial kernels.

We recall the lower bound instance: the noise distribution is $N(0, \sigma^2 Id)$ and the distribution for $Z$ is s.t. every coordinate is first chosen to be non-zero with probability $k/n$, and if it is non-zero, it's set as an independent sample from $N(0, \gamma^2)$. This construction makes $Z$ approximately $k$-sparse with high probability while making its coordinates independent. We choose $A$ as an arbitrary orthogonal matrix, so $m = n$. We choose $w$ to be an arbitrary $\pm 1$ sign vector, so $w_i^2 = 1$ for every $i$.

As a warmup, we first show that linear predictors, and subsequently fixed low degree polynomials cannot achieve the information-theoretic rate [12] of $O(\sigma^2 k \log n)$ – in fact, we will show that they achieve a "trivial" rate. Furthermore, we will show that even if the degree of our polynomials is growing with $n$, if $d = o(\log n / \log \log n)$ the state of affairs is similar.

## B.1    WARMUP: LINEAR PREDICTORS

As a warmup, and to illustrate the main ideas of the proof techniques, we first consider the case of linear predictors. (i.e. kernels of degree 1.)

The main idea is to use a bias-variance trade-off: namely, we show that the linear predictor we use, say $f(x) = \langle \tilde{w}, x \rangle$ either has to have too high of a variance (when $\|\tilde{w}\|$ is large), or otherwise has too high of a bias. (Recall, the bias captures how well the predictor captures the expectation.)

We prove:

**Theorem B.1.** *For any $\tilde{w} \in \mathbb{R}^n$,*

$$\mathbb{E}[(\langle \tilde{w}, X \rangle - Y)^2] \geq \gamma^2 k \frac{\sigma^2}{\gamma^2 (k/n) + \sigma^2}$$

Before giving the proof, let us see how the theorem should be interpreted.

The trivial estimator which always returns 0 makes error of order $\gamma^2 K$ and a good estimator (such as thresholding) should instead make error of order $\sigma^2 K \log n$ when $\gamma >> \sigma \sqrt{\log n}$. The next theorem shows that as long as the signal to noise ratio is not *too high*, more specifically as long as $\gamma^2 (k/n) = o(\sigma^2)$, any linear estimator must make square loss of $\Omega(\gamma^2 k)$, i.e. not significantly better than the trivial 0 estimate.

Note that the most interesting (and difficult) regime is when the signal is not too much larger than the noise, e.g. $\gamma^2 = \sigma^2 \text{polylog}(n)$ in which case it is definitely true that $\gamma^2 (k/n) << \sigma^2$.

*Proof.* Note that

$$\langle \tilde{w}, x \rangle - y = \langle \tilde{w}, Az + \xi \rangle - \langle w, z \rangle = \langle A^\top \tilde{w} - w, z \rangle + \langle \tilde{w}, \xi \rangle$$

which gives the following *bias-variance decomposition* for the square loss:

$$\begin{aligned}
\mathbb{E}[(\langle \tilde{w}, x \rangle - y)^2] &= \mathbb{E}[(\langle A^\top \tilde{w} - w, z \rangle + \langle \tilde{w}, \xi \rangle)^2] \\
&= \mathbb{E}[\langle A^\top \tilde{w} - w, z \rangle^2 + \langle \tilde{w}, \xi \rangle^2] \\
&= \frac{k}{n} \gamma^2 \|A^\top \tilde{w} - w\|_2^2 + \sigma^2 \|\tilde{w}\|_2^2 \\
&= \frac{k}{n} \gamma^2 \|\tilde{w} - Aw\|_2^2 + \sigma^2 \|\tilde{w}\|_2^2
\end{aligned}$$

where in the second-to-last step we used that the covariance matrix of $Z$ is $\gamma^2 (k/n) I$, and in the last step we used that $A$ is orthogonal. Now observe that if we fix $R = \|\tilde{w}\|_2$, then by the Pythagorean theorem the minimizer of the square loss is given by the projection of $Aw$ onto the $R$-dilated unit sphere, so $\tilde{w} = \sqrt{R^2/m}(Aw)$ since $\|Aw\|_2 = \|w\|_2 = \sqrt{m}$. In this case the square loss is then of the form

$$\frac{k}{n} \gamma^2 \|\sqrt{R^2/m}(Aw) - Aw\|_2^2 + \sigma^2 \|\tilde{w}\|_2^2 = \frac{k}{n} \gamma^2 (R - \sqrt{m})^2 + \sigma^2 R^2$$

---

[12]See Remark B.1 for why this is the optimal information-theoretic rate.

and the risk is minimized when

$$0 = 2\frac{k}{n}\gamma^2(R - \sqrt{m}) + 2\sigma^2 R$$

i.e. when

$$R = \frac{\gamma^2(k/n)}{\gamma^2(k/n) + \sigma^2}\sqrt{m}$$

so the minimum square loss is

$$(\sqrt{m} - R)\sigma^2 R + \sigma^2 R^2 = \sigma^2\frac{\gamma^2 k}{\gamma^2(k/n) + \sigma^2}$$

since $m = n$.

$\square$

## B.2 STRUCTURE OF THE OPTIMAL ESTIMATOR: PROOF OF LEMMA 5.1

*Proof of Lemma 5.1.* Let $X' = A^\top X$, so by orthogonality $X' = Z + \xi'$ where $\xi' \sim N(0, \sigma^2 Id)$. Observe that if we look at the optimum over all functions $f$, we see that

$$\min_f \mathbb{E}[(f(X') - \langle w, Z\rangle)^2] = \mathbb{E}[(\mathbb{E}[\langle w, Z\rangle|X'] - \langle w, Z\rangle)^2]$$

$$= \mathbb{E}[(\sum_i w_i\mathbb{E}[Z_i|X'] - \langle w, Z\rangle)^2]$$

$$= \mathbb{E}[(\sum_i w_i\mathbb{E}[Z_i|X_i'] - \langle w, Z\rangle)^2].$$

where where in the first step we used that the conditional expectation minimizes the squared loss, in the second step we used linearity of conditional expectation, and in the last step we used that $Z_i$ is independent of $X'_{\neq i}$.

By the Pythagorean theorem, the optimal degree $d$ polynomial $f_d^*$ is just the projection of $\sum_i w_i\mathbb{E}[Z_i|X_i']$ onto the space of degree $d$ polynomials. On the other hand observe that

$$\mathbb{E}[(\sum_i w_i\mathbb{E}[Z_i|X_i'] - \langle w, Z\rangle)^2] = \sum_i w_i^2\mathbb{E}[(\mathbb{E}[Z_i|X_i'] - Z_i)^2]$$

so the optimal projection $f_d^*$ is just $\sum_i w_i f_{i,d}^*(X_i')$ where $f_{i,d}^*$ is just the projection of each of the $\mathbb{E}[Z_i|X_i']$. Therefore $f_d^*$ has no mixed monomials. $\square$

**Remark B.1.** *The previous calculation shows additionally that the problem of minimizing the squared loss for predicting $Y$ is equivalent to that of minimizing the squared loss for the sparse regression problem of recovering $Z$. It is a well-known fact that the information theoretic rate for sparse regression (with our normalization convention) is $\Theta(\sigma^2 k)$ (see for example (Rigollet, 2017)), and so the information-theoretic rate for predicting $Y$ is the same, and is matched by Theorem A.1.*

## B.3 BIAS-VARIANCE, FOURIER ANALYSIS AND PROOF OF THEOREM 4.2

We recall that the lower bound for polynomials combines the observation of Lemma 5.1 with a bias-variance tradeoff calculation using Fourier analysis on orthogonal polynomials. Concretely, since the noise we chose for the lower bound instance is Gaussian, the most convenient basis will be the Hermite polynomals.

We recall the *probabilist's Hermite polynomial* $He_n(x)$, defined by the recurrence relation

$$He_{n+1}(x) = xHe_n(x) - nHe_{n-1}(x). \tag{1}$$

where $He_0(x) = 1, He_1(x) = x$. In terms of this, the *normalized Hermite polynomial* $H_n(x)$ is

$$H_n(x) = \frac{1}{\sqrt{n!}}He_n(x).$$

Let $H_{\mathbf{n}}(x)$ for a vector of indices $\mathbf{n} \in \mathbb{N}_0^m$ denote the multivariate polynomial $\Pi_{i=1}^m H_{\mathbf{n}_i}(x_i)$. It's easy to see the polynomials $H_{\mathbf{n}}(x)$ form an orthogonal basis with respect to the standard m-variate Gaussian distribution. As a consequence, we get

$$\mathbb{E}_{X \sim \mathcal{N}(0, \sigma^2 I)} H_{\mathbf{n}}(X/\sigma) H_{\mathbf{n}'}(X/\sigma) = \begin{cases} 0, & \text{if } \mathbf{n} \neq \mathbf{n}' \\ 1, & \text{otherwise} \end{cases}$$

which gives us Plancherel's theorem:

**Theorem B.2** (Plancherel in Hermite basis). *Let* $f(x) = \sum_{\mathbf{n}} \widehat{f}(\mathbf{n}) H_{\mathbf{n}}(x/\sigma)$*, then*

$$\mathbb{E}_{X \sim \mathcal{N}(0, \sigma^2 I)}[|f(X)|^2] = \sum_{\mathbf{n}} |\widehat{f}(\mathbf{n})|^2$$

We can use Plancherel's theorem to get lower bounds on the noise sensitivity of degree $d$ polynomials. This will be an analogue of the variance.

**Lemma B.1.** *[Variance analogue in Hermite basis] Let* $f(x) = \sum_{\mathbf{n}} \widehat{f}(\mathbf{n}) H_{\mathbf{n}}(x/\sigma)$ *and let* $f_{\neq 0} := f - \widehat{f}(0)$*. Then*

$$\mathbb{E}[(f(A^\top X) - Y)^2] \geq (1 - k/n) \|\widehat{f}_{\neq 0}\|_2^2$$

*Proof.* First we suppose $Z$ (and thus $Y$) is fixed and consider the randomness of the noise. Let $S$ denote the support of $Z$. Recall that $A^\top x = Z + \xi'$ where $\xi' \sim \mathcal{N}(0, \sigma^2 I_{n \times n})$. Define $f_Z(\xi) := f(Z + \xi) - Y$, then by Plancherel

$$\mathbb{E}_\xi[(f(A^\top x) - Y)^2] = \mathbb{E}_{\xi'}[f_Z(\xi')^2] = \sum_{\mathbf{n}} |\widehat{f_Z}(\mathbf{n})|^2$$

Furthermore

$$\sum_{\mathbf{n}} |\widehat{f_Z}(\mathbf{n})|^2 \geq \sum_{\mathbf{n}: supp(\mathbf{n}) \not\subset S} |\widehat{f}(\mathbf{n})|^2$$

because $(\xi' + Z)|_{S^c} = \xi'|_{S^c}$ so by expanding out $f_Z$ in terms of the fourier expansion of $f$, we see $\widehat{f_Z}(\mathbf{n}) = \widehat{f}(\mathbf{n})$ for $\mathbf{n}$ such that $supp(\mathbf{n}) \not\subset S$. Finally the probability $\mathbf{n} \subset S$ for $\mathbf{n} \neq 0$ is upper bounded by the probability a single element of its support is in $S$, which is $k/n$. Therefore

$$\mathbb{E}_{Z, \xi}[(f(A^\top x) - Y)^2] \geq \sum_{\mathbf{n}} |\widehat{f}(\mathbf{n})|^2 \mathbb{E}[\mathbb{1}_{supp(\mathbf{n}) \not\subset S}] \geq \sum_{\mathbf{n} \neq 0} (1 - k/n) |\widehat{f}(\mathbf{n})|^2$$

which proves the result. $\qquad\square$

Next we give a lower bound for the bias, showing that if $\|\widehat{f}_{\neq 0}\|_2^2$ is small for a low-degree polynomial, it cannot accurately predict $y$. Here we will assume $f$ is of the form given by Lemma 5.1.

**Lemma B.2** (Low variance implies high bias). *Suppose* $f$ *is a multivariate polynomial of degree* $d$ *with no mixed monomials, i.e.* $f(x) = \sum_i f_i(x_i)$ *where* $f_i$ *is a univariate polynomial of degree* $d$*. Expand* $f$ *in terms of Hermite polynomials as* $f(x) = \sum_{\mathbf{n}} \widehat{f}(\mathbf{n}) H_{\mathbf{n}}(x/\sigma)$*. Then*

$$\mathbb{E}[(f(A^\top X) - Y)^2] \geq (k/n) \sum_{i=1}^n w_i^2 \max(0, \gamma - \sqrt{\sum_{i=1}^n |\hat{f}(ke_i)|^2 (d+1)^{3d+2} (1 + (\gamma/\sigma)^d)})^2$$

Before proving the lemma, let us see how it proves the main theorem:

*Proof of Theorem 4.2.* By Lemma 5.1, Lemma B.1, and Lemma B.2 we have that for the $f$ which minimizes the square loss among degree $d$ polynomials, we have a variance-type lower bound

$$\mathbb{E}[(f(A^\top X) - Y)^2] \geq (1 - k/n) \sum_{i=1}^n \sum_{k=1}^d |\hat{f}(ke_i)|^2$$

and (using that $w_i^2 = 1$ to simplify) a bias-type lower bound

$$\mathbb{E}[(f(A^\top X) - Y)^2] \geq (k/n) \sum_{i=1}^{n} \max(0, \gamma - \sqrt{\sum_{i=1}^{n} |\hat{f}(ke_i)|^2 (d+1)^{3d+2}(1 + (\gamma/\sigma)^d))^2}.$$

Let $\|\hat{f}_i\|_2 := \sqrt{\sum_{i=1}^{n} |\hat{f}(ke_i)|^2}$. Then averaging these lower bounds and simplifying using $k < n/2$ gives

$$\mathbb{E}[(f(A^\top X) - Y)^2] \geq (1/4) \sum_{i=1}^{n} \max(\|\hat{f}_i\|_2, \sqrt{k/n}(\gamma - \|\hat{f}_i\|_2 (d+1)^{3d+2}(1 + (\gamma/\sigma)^d)))^2$$

$$\geq (1/4) \sum_{i=1}^{n} \frac{\gamma^2 (k/n)}{(1 + \sqrt{k/n}(d+1)^{3d+2}(1 + (\gamma/\sigma)^d))^2}$$

$$\geq (1/4) \frac{\gamma^2 k}{(1 + \sqrt{k/n}(d+1)^{3d+2}(1 + (\gamma/\sigma)^d))^2}$$

$\square$

Returning to the proof of Lemma B.2, we have:

*Proof of Lemma B.2.* Since $f$ has no mixed monomials, we get for the Hermite expansion that $\hat{f}(\mathbf{n}) = 0$ unless $|supp(\mathbf{n})| \leq 1$. Let $X' := A^\top X = Z + \xi'$ where $\xi' \sim N(0, \sigma^2 I)$. Next observe by independence that

$$\mathbb{E}[(f(X') - Y)^2] = \sum_i w_i^2 \mathbb{E}[(f_i(X_i') - Z_i)^2] \geq (k/n) \sum_i w_i^2 \mathbb{E}[(f_i(X_i') - Z_i)^2 | Z_i \neq 0]$$

where the last inequality follows since there is a $k/n$ chance that $Z_i \sim N(0, \sigma^2 I)$, equivalently that $Z_i \neq 0$. By the conditional Jensen's inequality we have

$$(k/n) \sum_i w_i^2 \mathbb{E}[(f_i(X_i') - Z_i)^2 | Z_i \neq 0] \geq (k/n) \sum_i w_i^2 \mathbb{E}[(\mathbb{E}[f_i(X_i')|Z_i] - Z_i)^2 | Z_i \neq 0].$$

Observe that $f_i(X_i') = \sum_{k=0}^{d} \hat{f}(ke_i) H_k(Z_i/\sigma + \xi/\sigma)$ and let $g_i(Z_i) := \mathbb{E}[f_i(X_i')|Z_i] - Z_i$, so then $g_i$ is a polynomial of degree $d$ in $Z_i$. Write the Hermite polynomial expansion of $g_i$ in terms of $H_k(Z_i/\gamma)$ as

$$g_i(x) = \sum_{k=0}^{d} \hat{g}_i(k) H_k(Z_i/\gamma),$$

then by Plancherel's formula

$$(k/n) \sum_i w_i^2 \mathbb{E}[(\mathbb{E}[g(Z_i) - Z_i)^2 | Z_i \neq 0] = (k/n) \sum_i w_i^2 \sum_{k=0}^{d} |\hat{g}_i(k)|^2 \geq (k/n) \sum_i w_i^2 |\hat{g}_i(1)|^2$$

and it remains to lower bound $|\hat{g}_i(1)|$. By orthogonality and direct computation,

$$\hat{g}_i(1) = \mathbb{E}_{Z_i \sim N(0,\gamma)}[(\mathbb{E}[f_i(X_i')|Z_i] - Z_i) H_1(Z_i/\gamma)] = -\gamma + \mathbb{E}_{Z_i \sim N(0,\gamma)}[\mathbb{E}[f_i(X_i')|Z_i](Z_i/\gamma)].$$

Now we upper bound the last term

$$\mathbb{E}_{Z_i \sim N(0,\gamma)}[\mathbb{E}[f_i(X_i')|Z_i](Z_i/\gamma)] = \hat{f}(0)\mathbb{E}[Z_i/\gamma] + \sum_{k=1}^{d} \hat{f}(ke_i)\mathbb{E}_{Z_i \sim N(0,\gamma)}[\mathbb{E}[H_k(Z_i/\sigma + \xi'/\sigma)|Z_i](Z_i/\gamma)]$$

$$= \sum_{k=1}^{d} \hat{f}(ke_i)\mathbb{E}_{Z_i \sim N(0,\gamma)}[H_k(Z_i/\sigma + \xi'/\sigma)(Z_i/\gamma)]$$

$$\leq \left(\sum_{k=1}^{d} |\hat{f}(ke_i)|^2\right)^{1/2} \left(\sum_{k=1}^{d} \mathbb{E}_{Z_i \sim N(0,\gamma)}[H_k(Z_i/\sigma + \xi'/\sigma)(Z_i/\gamma)]^2\right)^{1/2}$$

where the second equality is by the law of total expectation and the last inequality is Cauchy-Schwarz. Using the recurrence relation (1), we can bound the sum of the absolute value of the coefficients of $H_k(x)$ by $k^k/\sqrt{k!} \leq k^k$. We can also bound the moments of the absolute value of a Gaussian by $\mathbb{E}_{\xi \sim N(0,1)}[|\xi|^k] \leq k^k$. Therefore by Holder's inequality

$$
\begin{aligned}
\mathbb{E}_{Z_i \sim N(0,\gamma)}[H_k(Z_i/\sigma + \xi'/\sigma)(Z_i/\gamma)] &\leq k^k \sup_{\ell=1}^k |\mathbb{E}_{Z_i \sim N(0,\gamma)}[(Z_i/\sigma + \xi'/\sigma)^\ell (Z_i/\gamma)]| \\
&\leq k^k \sup_{\ell=1}^k \mathbb{E}_{Z_i \sim N(0,\gamma)}[(|Z_i|/\sigma + |\xi'|/\sigma)^\ell (|Z_i|/\gamma)] \\
&\leq 2^k k^k \sup_{\ell=1}^k (\mathbb{E}_{Z_i \sim N(0,\gamma)}[|Z_i|^{\ell+1}/\sigma^\ell \gamma] + \mathbb{E}_{Z_i \sim N(0,\gamma)}[|\xi'|^\ell |Z_i|/\sigma^\ell \gamma)]) \\
&\leq 2^k k^k [\max(1, (\gamma/\sigma)^k)(k+1)^{k+1} + k^k] \\
&\leq (k+1)^{3k+1}(1 + (\gamma/\sigma)^k).
\end{aligned}
$$

Therefore by reverse triangle inequality

$$
|\hat{g}_i(1)|^2 \geq \max\left(0, \gamma - \sqrt{\sum_{i=1}^n |\hat{f}(ke_i)|^2 (d+1)^{3d+2}(1+(\gamma/\sigma)^d))^2}\right).
$$

$\square$

**Remark B.2.** *Remarks on results:*

We make a few remarks regarding the results in this section. Recall that $\gamma^2 k$ is the square loss of the trivial zero-estimator. Suppose as before that $\gamma = \Theta(\sigma^2 polylog(n))$, then we see that if $d = o(\log n/\log\log n)$ then the denominator of the lower bound tends to 1, hence any such polynomial estimator has a rate no better than that of the trivial zero-estimate.

It is possible to derive a similar statement to Theorem 4.2 that holds with high probability instead of in expectation for polynomials of degree $o(\log n/\log\log n)$. All that is needed is to bound the contribution to the expectation from very rare tail events when the realization of the noise $\xi$ is atypically large. Since the polynomials we consider are very low degree $o(\log n/\log\log n)$, they can only grow at a rate of $x^d = x^{o(\log(n)/\log\log n)}$; thus standard growth rate estimates (e.g. the Remez inequality) combined with the Gaussian tails of the noise can be used to show that a polynomial which behaves reasonably in the high-probability region (e.g. which has small w.h.p. error) cannot contribute a large amount to the expectation in the tail region.

## C  UPPER BOUNDS FOR POLYNOMIAL KERNELS

In this section, we construct polynomials achieving close to the information-theoretic optimal rate of degree only $O(\log^2 m)$. Recall this is nearly optimal due to our previous lower bound of $\Omega(\log n)$.

As previously mentioned, the key technical result here will be Theorem 5.2, giving the construction of a new polynomial approximation to ReLU. Before proceeding to the proof of that theorem, we show how it implies the final result, Theorem 4.3.

Towards that, we substitute our polynomial construction for $\rho_\tau$ into our ReLU neural network and derive the analogous version of Lemma A.2. First, define $M_\tau = M + 2\tau$ and let

$$
\tilde{\rho}_{d,\tau,M} = p_{d,\tau/M_\tau,M_\tau}(x - \tau) + p_{d,\tau/M_\tau,M_\tau}(-x + \tau)
$$

where $p$ is the polynomial constructed in Theorem 5.2. We then have:

**Lemma C.1.** *Suppose $\epsilon, \tau > 0$ and $M \geq 1$. Then for all $d \geq d_0 = \Omega(\frac{M_\tau}{\tau} \log^2(\frac{M_\tau}{\epsilon\tau}))$, for $|x| \in (\tau, M_\tau)$ we have*

$$
|\tilde{\rho}_{d,\tau,M}(x) - x| \leq 3\tau + \epsilon
$$

*and for $|x| \leq \tau$ we have*

$$
|\tilde{\rho}_{d,\tau,M}(x)| \leq \epsilon
$$

*Proof.* By the guarantee of Theorem 5.2, we see that for for $|x| \le \tau$ that

$$|\tilde{\rho}_{d,\tau,M}(x)| \le 28 M_\tau \sqrt{\frac{dM_\tau}{\tau\pi}} e^{-\sqrt{\pi\tau d/4M_\tau}}.$$

Thus we see that taking $d = \Omega(\frac{M_\tau}{\tau} \log^2(\frac{M_\tau}{\epsilon\tau}))$ suffices to make the latter expression at most $\epsilon$. Similarly for $|x| > \tau$ we know that

$$|\tilde{\rho}_{d,\tau,M}(x)| \le 2\tau + 2M_\tau \sqrt{\frac{4\tau}{M_\tau\pi d}} + 26 M_\tau \sqrt{\frac{dM_\tau}{\tau\pi}} e^{-\sqrt{\pi\tau d/4M_\tau}}$$

and taking $d = \Omega(\frac{M_\tau}{\tau} \log^2(\frac{M_\tau}{\epsilon\tau}))$ with sufficiently large constant guarantees the middle term is at most $\tau$ and the last term is at most $\epsilon$. $\qquad\square$

Using this, we can show that if we use a polynomial of degree $\Omega((M/\sigma\sqrt{\log n}) \log^2 m)$ we can achieve similar statistical performance to the ReLu network:

**Lemma C.2.** *Suppose $A$ is $\mu$-incoherent i.e. $\|A^\top A - Id\|_\infty \le \mu$. Let $z$ be an arbitrary fixed vector such that $\|z\|_1 \le M$ and $|supp(z)| \le k$. Suppose $x = Az + \xi$ where $\xi \sim N(0, \sigma^2 Id_{n\times n})$. Then for some $\tau = \Theta(\sigma\sqrt{(1+\mu)\log m} + \mu M)$, for any $d \ge d_0 = \Omega(\frac{M_\tau}{\tau} \log^2(M_\tau m/\tau^2))$, if we take $\hat{z} := \tilde{\rho}_{d,\tau,M}^{\otimes n}(A^\top x)$, then with high probability we have $\|\hat{z} - z\|_1 \le 6k\tau$.*

*Proof.* Apply Lemma C.1 with $\epsilon = \tau/m$. Then we see for $|x| \in (\tau, M_\tau)$ we havn

$$|\tilde{\rho}_{d,\tau,M}(x) - x| \le (3 + 1/m)\tau \le 4\tau$$

and for $|x| \le \tau$ we have

$$|\tilde{\rho}_{d,\tau,M}(x)| \le \tau/m$$

Observe that

$$A^\top x = z + (A^\top A - Id)z + A^\top \xi.$$

Note that entry $i$ of $A^\top \xi$ is $\langle A_i, \xi \rangle$ where $\|A_i\|_2^2 \le (1+\mu)$ so $(A^t \xi)_i$ is Gaussian with variance at most $\sigma^2(1+\mu)$.

By choosing $\tau$ with sufficiently large constant, then applying the sub-Gaussian tail bound and union bound, with high probability all coordinates not in the true support are thresholded to at most $\tau/m$. Similarly we see that for each of the coordinates in the support, an error of at most $5\tau$ is made. Therefore $\|\hat{z} - z\|_1 \le 5k\tau + m(\tau/m) \le 6k\tau$. $\qquad\square$

Now we have all the ingredients to prove Theorem 4.3:

*Proof of Theorem 4.3.* Define an estimate for $Y$ by taking $\hat{Z}_{d,M} := \tilde{\rho}_{d,\tau,M}^{\otimes n}(A^\top X)$ where $\tau$ is defined as in the Lemma, and then taking $\hat{Y}_{d,M} := \langle w, \hat{Z}_{d,M} \rangle$. Applying the previous Lemma, we get analogous versions of Theorem 4.1 by the same argument as in that theorem. $\qquad\square$

Finally, we return to the proof of the key Theorem 5.2:

*Proof of Theorem 5.2.* We start with the case where $R = 1/2$. We build the approximation in two steps. First we approximate ReLu by the following "annealed" version of ReLu, for parameters $\beta > \pi, \tau > 0$ to be optimized later:

$$g_\beta(x) = \frac{1}{\beta} log(1 + e^{\beta x})$$

$$f_{\beta,\tau}(x) = g_\beta(x - \tau).$$

Observe that when we look at negative inputs, $g_\beta(-x) = \frac{1}{\beta} \log(1 + e^{-\beta x}) \le \frac{1}{\beta} e^{-\beta x}$. Therefore when $x < 0$, $f_\beta(x) \le \frac{1}{\beta} e^{-\beta\tau}$.

For the second step,, we need to show $f_\beta$ can be well-approximated by low-degree polynomials. In fact, because $f_\beta$ is analytic in a neighborhood of the origin, it turns out that its optimal rate of

approximation is determined exactly by its complex-analytic properties. More precisely, define $D_\rho$ to be the region bounded by the ellipse in $\mathbb{C} = \mathbb{R}^2$ centered at the origin with equation

$$\frac{x^2}{a^2} + \frac{y^2}{b^2} = 1$$

with semi-axes $a = \frac{1}{2}(\rho + \rho^{-1})$ and $b = \frac{1}{2}|\rho - \rho^{-1}|$; the focii of the ellipse are $\pm 1$. For an arbitrary function $f : [-1, 1] \to \mathbb{R}$, let $E_d(f)$ denote the error of the best polynomial approximation of degree $d$ in infinity norm on the interval $[-1, 1]$ of $f$. Then the following theorem of Bernstein exactly characterizes the growth rate of $E_d(f)$:

**Theorem C.1** (Theorem 7.8.1, (DeVore and Lorentz, 1993)). *Let $f$ be a function defined on $[-1, 1]$. Let $\rho_0$ be the supremum of all $\rho$ such that $f$ has an analytic extension on $D_\rho$. Then*

$$\limsup_{d \to \infty} \sqrt[d]{E_d(f)} = \frac{1}{\rho_0}$$

For our application we need only the upper bound and we need a quantitative estimate for finite $n$. Following the proof of the upper bound in (DeVore and Lorentz, 1993), we get the following result:

**Theorem C.2.** *Suppose $f$ is analytic on the interior of $D_{\rho_1}$ and $|f(z)| \leq M$ on the closure of $D_{\rho_1}$. Then*

$$E_d(f) \leq \frac{2M}{\rho_1 - 1}\rho_1^{-n}$$

The proof is fairly simple: by writing $f$ in terms of $\cos(x)$ one gets an expansion into Chebyshev polynomials and it suffices to bound the coefficients of the corresponding Fourier series: to do this, we write them as integrals over the unit circle, and use the analyticity assumption on $D_{\rho_1}$ to contour shift the integral to a different circle, which immediately gives us the desired exponential decay. For details see (DeVore and Lorentz, 1993).

We will now apply this theorem to $g_\beta$. First, we claim that $g_\beta$ is analytic on $D_{\rho_1}$ where $\rho_1$ is the solution to this equation for the semi-axis of the ellipse:

$$\frac{1}{2}(\rho - \rho^{-1}) = \frac{\pi}{2\beta}$$

which is

$$\rho_1 = \frac{\sqrt{4\beta^2 + \pi^2} + \pi}{2\beta} > 1 + \pi/2\beta.$$

To see this, first extend $\log$ to the complex plane by taking a branch cut at $(-\infty, 0]$. To prove $g_\beta$ is analytic on $D_{\rho_1}$, we just need to prove that $1 + e^{\beta z}$ avoids $(-\infty, 0]$ for $z \in D_{\rho_1}$. This follows because by the definition of $\rho_1$, for every $z \in D_{\rho_1}$, $\Im(z) < \frac{\pi}{2\beta}$ hence $\Re(1 + e^{\beta z}) \geq 1$. We also see that for $z \in D_{\rho_1}$,

$$|g_\beta(z)| = \frac{1}{\beta}|\log(1 + e^{\beta z})| \leq \frac{1}{\beta} \sup_{w \in D_{\beta\rho_1}} |\log(1 + e^w)| \leq \frac{1}{\beta}(\log(1 + e^\beta) + \pi) < 6.$$

Therefore by Theorem C.2 we have

$$E_d(g_\beta) \leq \frac{12\beta}{\pi}(1 + \pi/2\beta)^{-n} \leq \frac{12\beta}{\pi}e^{-\pi n/4\beta}$$

where in the last step we used that $1 + x \geq \exp(x/2)$ for $x < 1/2$ and that $\beta > \pi$. Let $\tilde{g}_{\beta,d}$ denote the best polynomial approximation to $g_\beta$ of degree $d$ and let $\tilde{f}_{\beta,\tau,d} = \tilde{g}_{\beta,d}(x - \tau)$

Thus for $x \in [-1 + \tau, 0]$,

$$|ReLu(x) - \tilde{f}_{\beta,\tau,d}(x)| \leq |f_{\beta,\tau}(x)| + |\tilde{g}_{\beta,d}(x - \tau) - g_{\beta,\tau}(x - \tau)| \leq \frac{1}{\beta}e^{-\beta\tau} + \frac{12\beta}{\pi}e^{-\pi d/4\beta}$$

Take $\beta = \sqrt{\pi d/4\tau}$ and require $d > 7$ so that $\beta > 1$, then for $x \in [-1 + \tau, 0]$,

$$|ReLu(x) - \tilde{f}_{\beta,\tau,d}(x)| \leq 7\sqrt{\frac{d}{\tau\pi}}e^{-\sqrt{\pi\tau d/4}}$$

For $x \in (0, 1 - \tau]$ we have by the 1-Lipschitz property of $g_\beta$ and calculus that

$$|x - f_{\beta,\tau}(x)| \leq \tau + |x - g_\beta(x)| \leq \tau + \frac{\log 2}{\beta}$$

so

$$|ReLu(x) - \tilde{f}_{\beta,\tau,d}(x)| \leq |x - f_{\beta,\tau}(x)| + |\tilde{g}_{\beta,d}(x - \tau) - g_{\beta,\tau}(x - \tau)| \leq \tau + \frac{\log 2}{\beta} + \frac{12\beta}{\pi} e^{-\pi d/4\beta}.$$

Plugging in our value of $\beta$ and using $\log 2 \leq 1$ gives

$$|ReLu(x) - \tilde{f}_{\beta,\tau,d}(x)| \leq \tau + \sqrt{\frac{4\tau}{\pi d}} + 6\sqrt{\frac{d}{\tau\pi}} e^{-\sqrt{\pi\tau d/4}}$$

Now the result for general $R$ follows by taking $p_d(x) = 2R\tilde{f}_{\beta,\tau,d}(x/2R)$, since $2R \cdot ReLu(x/2R) = ReLu(x)$ and $[-1/2, 1/2] \subset [-1 + \tau, 1 - \tau]$. $\qquad\square$

