# OpenReview forum: "The Comparative Power of ReLU Networks and Polynomial Kernels in the Presence of Sparse Latent Structure"
_ICLR.cc/2019/Conference_

### Official Review · AnonReviewer1 · 2018-10-26
**The Comparative Power of ReLU Networks and Polynomial Kernels in the Presence of Sparse Latent Structure**

**Rating:** 7
**Confidence:** 3

**Review:**

In this paper, authors analyze the performance of neural networks and polynomial kernels in a natural regression setting where the data enjoys sparse latent structure, and the labels depend in a simple way on the latent variables. They give an almost-tight theoretical analysis of the performance and verify them with simulations.

Authors motivated the theoretical analysis from typical applications, for which the desired function can be only important to be approximated well on the relevant part of domains. Instead of formalizing the above problem, authors tackle a particular simple question. However, it is not easy to understand the relationships between the two problems.

A regression task is studied where the data has a sparser latent structure. Authors measure the performance of estimators via the expected reconstruction error from theoretical perspectives for both two-layer ReLU network and polynomial kernel. Empirical experiments will be even better to show the performance of some applications consistent with the theoretical results.

---

### Official Review · AnonReviewer2 · 2018-11-02
**Interesting "Relevant domain" based approximation of ReLU to approximate sparse latent structures**

**Rating:** 7
**Confidence:** 3

**Review:**

The paper studies the representational power of two-layer ReLU networks and polynomials for approximating a linear generative model for data with sparsity in the latent vector. They show that ReLU networks achieve optimal rate whereas low degree polynomials get a much worse rate.

Overall, the results are strong, the authors provide a lower bound on the degree of polynomial needed to approximate the model indicating the power of non-linearity. The observation of moving away from uniform approximators is well-motivated. The approximation theorem for ReLU is intriguing and uses new ideas which I have not seen before and are potentially useful in other applications. So far, only rational functions have been able to give such approximation guarantees. However, the motivation for studying sparse linear regression from a representation view-point is not very clear. Ideally, you would like to study representation for more complex models.

Questions/Comments:
- Related work is missing prior work at the intersection of kernel methods and neural networks, please update.
- Define notation before using, for example, \rho_\tau^{⨂m}
- Expand proof sketches, they are not very clear, also full proofs are written with not much detail.
- Is the dependence on \mu tight? The current dependence sort of suggests that you need the observation matrix to be very close to identity.
- Proof of Lemma B.1 is unclear, could you explain how you deduce the lemma from the inequality?

---

### Official Review · AnonReviewer3 · 2018-11-04

**Rating:** 7
**Confidence:** 3

**Review:**

This paper studies the problem of understanding the representation power of neural nets with Relu activations for representing structured data. In order to formalize this, the authors consider data generated from a sparse generative model as follows: A sparse m-dimensional vector Z is sampled from a distribution over sparse vectors. In input X is formed
as AZ, where A is an incoherent matrix. The corresponding output is Y= w. X. The goal is to fit the data of the form (X_i, Y_i). The main result of the paper is that a 2-layer ReLU network can fit the data with near optimal error. On the other hand, low degree polynomials~(of degree up to log m) cannot fit the data with non-trivial error. Finally,
the authors also show that polynomials of degree polylog(m) can, in fact, fit the data as well as a 2-layer ReLU network. The paper is well written and provides new insights into the representation power of neural nets. It is also nice to know that ReLU networks can be approximated by low degree polynomials in the non-worst case scenario. This
is a good paper and I recommend acceptance.

---

### Author Response · Authors · 2018-11-24
**Response to Reviewers**

We thank the reviewers for their valuable feedback, which we have incorporated into the new revision of the paper. In particular, in response to AnonReviewer2, we added a discussion of a related work by Zhang et al. on kernel methods simulating neural networks and have added more details to both the proofs and proof sketches.

AnonReviewer2 and AnonReviewer1 asked about more complex models: we agree that sparse regression is a relatively simple model and that it would be nice to study more complex models as well. Since latent sparsity is a common feature in many models, this seemed like the natural place to start -- we hope the analysis of more sophisticated models will follow.

AnonReviewer2 also asked about tightness of the dependence on \mu: for noisy sparse linear regression, the polynomial dependence on \mu cannot be significantly improved due to issues of computational complexity. For instance, https://arxiv.org/pdf/1402.1918.pdf show that sparse linear regression with a statistical rate better than polynomial in \mu is computationally hard.  If, for example, small-degree polynomials existed (with dependency better than polynomial in \mu), these computational hardness results would be violated.

---

### Meta-Review · Area_Chair1 · 2018-12-17
**Interesting new analysis of function approximation in the presence of sparse latent structure**

**Confidence:** 5
**Recommendation:** Accept (Poster)

**Metareview:**

This paper makes a substantial contribution to the understanding of the approximation ability of deep networks in comparison to classical approximation classes, such as polynomials.  Strong results are given that show fundamental advantages for neural network function approximators in the presence of a natural form of latent structure.  The analysis techniques required to achieve these results are novel and worth reporting to the community.  The reviewers are uniformly supportive.